# Research on the Medicinal Chemistry and Pharmacology of *Taxus* × *media*

**DOI:** 10.3390/ijms25115756

**Published:** 2024-05-25

**Authors:** Xinyu Gao, Ni Zhang, Weidong Xie

**Affiliations:** 1State Key Laboratory of Chemical Oncogenomics, Shenzhen International Graduate School, Tsinghua University, Shenzhen 518055, China; gao-xy23@mails.tsinghua.edu.cn (X.G.); zhangn22@mails.tsinghua.edu.cn (N.Z.); 2Shenzhen Key Laboratory of Health Science and Technology, Institute of Biopharmaceutical and Health, Shenzhen International Graduate School, Tsinghua University, Shenzhen 518055, China

**Keywords:** *Taxus × media*, cultivation, phytochemical constituents, pharmacological activity

## Abstract

*Taxus* × *media*, belonging to the genus *Taxus* of the Taxaceae family, is a unique hybrid plant derived from a natural crossbreeding between *Taxus cuspidata* and *Taxus baccata*. This distinctive hybrid variety inherits the superior traits of its parental species, exhibiting significant biological and medicinal values. This paper comprehensively analyzes *Taxus × media* from multiple dimensions, including its cultivation overview, chemical composition, and multifaceted applications in the medical field. In terms of chemical constituents, this study delves into the bioactive components abundant in *Taxus × media* and their pharmacological activities, highlighting the importance and value of these components, including paclitaxel, as the lead compounds in traditional medicine and modern drug development. Regarding its medicinal value, the article primarily discusses the potential applications of *Taxus × media* in combating tumors, antibacterial, anti-inflammatory, and antioxidant activities, and treating diabetes. By synthesizing clinical research and experimental data, the paper elucidates the potential and mechanisms of its primary active components in preventing and treating these diseases. In conclusion, *Taxus × media* demonstrates its unique value in biological research and tremendous potential in drug development.

## 1. Introduction

*Taxus × media* (*Taxus × media* Rehder), a member of the genus *Taxus* in the family Taxaceae, is a plant that combines medicinal, timber, and ornamental values, thus possessing significant economic and research importance [1]. As a species within the Taxus genus, it plays a crucial role in the ecosystem and is highly valued in the field of medicine due to its rich bioactive compounds [2].

A thorough understanding of the natural habitat and cultivation of *Taxus × media* is vital for conserving this precious species and developing its medicinal resources. The natural distribution and ecological habits of *Taxus × media* provide a theoretical basis for biodiversity and ecosystem integrity. This species primarily grows in subtropical mountain forests, and its unique growth environment and ecological characteristics are crucial in formulating effective cultivation and conservation strategies [3]. With the increasing depletion of natural resources and environmental pressures, *Taxus × media* faces numerous challenges, including habitat loss, overharvesting, and the impact of climate change. This species is particularly susceptible to temperature fluctuations, which can affect the metabolic processes essential for synthesizing key bioactive compounds. Research has shown that under varying temperature conditions, *Taxus × media* expresses different proteins related to the synthesis of paclitaxel and other secondary metabolites. These proteins are involved in the precursor supply for the paclitaxel biosynthesis pathway, such as 1-deoxy-D-xylulose-5-phosphate synthase (DXS) and 1-deoxy-D-xylulose-5-phosphate reductoisomerase (DXR). This differential expression may explain the variations in the content of bioactive compounds under different environmental conditions. Additionally, experiments have demonstrated that environmental factors, such as temperature, significantly influence the content of these bioactive compounds. For instance, the content of paclitaxel found in *Taxus × media* is more than five times that of *Taxus mairei*. Furthermore, changes in precipitation patterns and an increased frequency of extreme weather events can further stress natural populations [4]. Understanding these dynamics is crucial for developing effective conservation strategies to ensure the survival of *Taxus × media* under changing climatic conditions [5]. Therefore, a comprehensive understanding of its cultivation status, assessment of the survival of wild populations, and exploration of effective conservation methods are essential for the protection of *Taxus × media* and for maintaining biodiversity and ecological balance.

The phytochemical constituents of *Taxus × media* form the material basis and core of its medicinal value. Paclitaxel, its most famous compound, has been widely used clinically for its antitumor activities [6]. However, recent studies have revealed that *Taxus × media* contains various other bioactive compounds [7]. These compounds’ extraction, preparation, analysis, and biological activity assessment are current research hotspots. A deep understanding of the structure and function of these chemical components is crucial for developing new drugs and treatment strategies.

Pharmacological studies on *Taxus × media* are not limited to its anticancer effects but also include exploring its potential as an anti-inflammatory, antimicrobial, and in other fields [8,9,10]. These studies reveal the potential effects and mechanisms of action of *Taxus × media* compounds and provide a theoretical basis for new drug development. Integrating clinical trial and laboratory research data can enhance the understanding of these compounds’ pharmacological properties and future applications. After revealing its rich medicinal potential, the challenge becomes how to utilize these compounds sustainably. Addressing the scarcity of *Taxus × media* medicinal resources and conservation challenges, finding synthetic substitutes, optimizing extraction processes, or developing new biotechnological methods are essential to sustainable utilization.

This review aims to thoroughly explore various research directions of *Taxus × media*, including its natural origin, cultivation status, chemical constituents, pharmacological activities, and application prospects, providing a comprehensive reference for future medical research and development.

## 2. Origin and Cultivation

### 2.1. Origin and Geographical Distribution

Globally, plants of the Taxaceae family, commonly known as yews, primarily grow in Asia, North America, and Europe, with a total of approximately five genera and 23 species. In China, the *Taxus* genus is represented by four native species, one variety, and one introduced species, along with a hybrid. The native species are *Taxus chinensis* (Pilger.) Rehd, *Taxus wallichiana* Zucc., *Taxus cuspidata* Siebold and Zucc and *Taxus yunnanensis* W.C.Cheng and L.K.Fu. The variety is *Taxus mairei* (Lemée and H.Lév.) S.Y.Hu, while the introduced species is *Taxus × media* Rehder [11]. *Taxus × media* is a species within the *Taxus* genus of the Taxaceae family, discovered in 1918 by American scholars as an artificial hybrid between the female *Taxus cuspidata* and the male *Taxus baccata*. Originally native to North America, specifically Canada and the United States, this plant has now been introduced and cultivated in several countries around the world, including China, India, Argentina, and South Korea, with a primary distribution in the Asian region [12,13]. Figure 1 shows the geographical distribution of *Taxus × media* in China.

In the mid-1990s, China introduced *Taxus × media* from Canada. In 1995, the Sichuan Provincial Academy of Forestry first introduced it into Sichuan province. After nearly 15 years of elite breeding and cultivation trials, a new variety with a high paclitaxel content—Chuanlin *Taxus × media*—was developed, along with an efficient cultivation technology system for *Taxus × media*. According to authoritative institutions, the biological characteristics of the introduced *Taxus × media* have remained stable without any mutations. It is now cultivated in various locations in China, including the Sichuan, Guangxi, and Shandong provinces [14].

Through selective breeding of *Taxus × media*, more than ten cultivars have been developed to date. The cultivation of this plant is primarily concentrated in areas with suitable altitude and climatic conditions, preferring deep, loose, slightly acidic sandy loam soils. In Taiwan, *Taxus × media* is a rare plant found in only six to seven locations. In the Himalayan region, it is typically found at altitudes between 1200 and 2000 m, while in Eastern China, most sites are below 1200 m. In Southern Vietnam, *Taxus × media* exists in several small sub-populations, such as in the Lam Dong and Khanh Hoa provinces [14].

*Taxus × media* is one of the yew species approved by the U.S. FDA for the extraction of paclitaxel [15]. Artificial cultivation has become an important sustainable development strategy due to the high demand for medicinal components such as paclitaxel in *Taxus × media* in recent years [16]. In several provinces of China, particularly in its natural distribution areas, numerous artificial cultivation bases of *Taxus × media* have been established. These bases alleviate the harvesting pressure on wild yew populations and provide opportunities to study their growth characteristics and medicinal value. Furthermore, the Chinese government and environmental organizations are implementing conservation measures to protect this precious species, including establishing protected areas and restricting commercial harvesting activities.

Prior to the pressing challenges posed by climate change, the *Taxus* species has faced significant threats due to human activities, primarily from overharvesting for their valuable medicinal compounds and habitat destruction for agricultural expansion. With the onset of climate change, these species are now confronting additional pressures such as increased temperatures and altered precipitation patterns, which can lead to physiological stress and reduced reproductive success [17]. These climatic factors can significantly shift the phenology and distribution of *Taxus* species, potentially leading to mismatches in the ecosystem interactions that sustain them. Moreover, the increased frequency of extreme weather events, such as droughts and heavy rains, can further destabilize their fragile habitats, leading to a higher risk of population decline. In light of these challenges, conservation efforts must not only address immediate threats from human exploitation but also incorporate adaptive strategies to mitigate the impact of climate change [18,19]. This includes the establishment of genetic reservoirs and assisted migration to areas predicted to remain climatically stable. Understanding the full scope of these threats is crucial to developing effective conservation strategies that ensure the long-term survival of the *Taxus* species in their natural habitats [5].

### 2.2. Artificial Cultivation and Comparative Analysis

Artificial cultivation of *Taxus × media* has become crucial due to the soaring demand for taxanes like paclitaxel, known for its anticancer properties. Optimized cultivation systems, which include controlled environments like greenhouses and open-field plantations, are tailored to enhance growth by modifying factors such as watering schedules, nutrient application, and pest management strategies. These systems crucially influence the plant’s secondary metabolite profile by allowing precise control over environmental conditions like light intensity, temperature, and soil pH. Studies, such as those exploring the synergistic effects of cyclodextrins and methyl jasmonate, have shown that such controlled environments can lead to an increase in taxane yield by up to 30% compared to traditional cultivation methods [20].

While existing research on the impact of climate and geographical factors on *Taxus × media* is limited, comparative studies within the Taxaceae family suggest that these factors significantly affect metabolite synthesis in similar species. For example, variations in temperature and light exposure were found to alter the concentration of baccatin III, a precursor to paclitaxel, by as much as 25% [21]. This indicates potential areas for future research, where exploring how specific environmental adjustments affect *Taxus × media* could lead to the development of more efficient cultivation strategies.

Comparative analysis between the wild and cultivated *Taxus × media* has revealed significant differences in their morphological and chemical profiles. Cultivated plants often exhibit higher growth rates and denser foliage, which are believed to contribute to their enhanced metabolite production. However, there are trade-offs, as these plants sometimes show lower diversity in certain secondary metabolites, which could affect their overall medicinal value. Advancing our understanding of these differences is essential for optimizing cultivation practices and ensuring the sustainability of *Taxus × media* resources [22].

## 3. Phytochemical Constituents

*Taxus × media*, a plant of significant medicinal value, has been a focal point of research in the fields of medicinal chemistry and botany [23]. The whole plant of *Taxus × media* is rich in phytochemical constituents, primarily comprising paclitaxel and its derivatives, various alkaloids, and flavonoids [24]. Analysis of the different medicinal parts of *Taxus × media* (such as bark, fresh leaves, seeds, and seed coat) has identified over 800 compounds, covering 11 subcategories [25], with their main components and contents presented in Table 1.

### 3.1. Chemical Composition of Different Parts of Taxus × media

Different parts of *Taxus × media* contain distinct phytochemical components. Using targeted metabolomics, studies have shown that tissues such as bark, fresh leaves, seeds, and receptacles contain a variety of substances, with different parts potentially housing unique compounds [2]. Figure 2 shows the representative phytochemical constituents of different parts of *Taxus × media*. 

#### 3.1.1. Chemical Composition of *Taxus × media* Leaves and Twigs

*Taxus × media* is an evergreen shrub which has abundant leaves and twigs that are easily harvested. Studies reveal significant differences in compound distribution among different tissues of *Taxus × media*, particularly in fresh leaves where taxane compounds are most abundant, resulting in a higher overall concentration of these components [2]. Additionally, the leaves and twigs of *Taxus × media* contain other chemical components such as flavonoids, volatile oils, and inositol [2,36].

The leaves and twigs of *Taxus × media* are rich in a variety of taxane components, including paclitaxel, cephalomannine, 10-deacetylbaccatin III (10-DAB) as the representative components [37]. Moreover, other diterpenoids have been identified, including 7,9-deacetyltaxinine, 9-deacetyltaxinine A, d-deacetyltaxinine B, taxinine-11, 2-deacetoxytaxinine E, 2-deacetoxytaxuspine C, taxagifin, and 12-oxide [38]. Among these, 10-DAB is the most abundant at approximately 0.75 mg/g, followed by paclitaxel, cephalomannine, and 10-deacetyl paclitaxel (10-DAT) at concentrations of approximately 0.6 mg/g, 0.5 mg/g, and 0.126 mg/g, respectively [39]. Further studies have shown that in *Taxus × media* of 15 years old, the paclitaxel content in one-year-old leaves and twigs can reach up to 0.02–0.04%, and even higher 10-DAB content at 0.08–0.15% [40,41].

The leaves and twigs are also rich in flavonoid compounds, including kaempferol, aromadendrin, apigenin, sciadopitysin, ginkgetin, luteolin, quercetin, amentoflavone, and others [38,42]. Phenolic compounds are also abundant, containing 4-hydroxy-benzaldehyde, p-hydroxybenzoic acid, and pyrocatechol [38]. The total flavonoid extraction from *Taxus × media* leaves and twigs is about 128.1 mg/g of dry weight [43].

Polysaccharides are present in varying amounts in the leaves and twigs, ranging from 1.1781 to 3.0115%, with an average content of 2.1367% [44]. The molecular weight of these polysaccharides is about 59.2 kilodaltons (kD), composed of approximately 365 sugar residues, with a ratio of rhamnose, arabinose, mannose, glucose, and galactose at approximately 4:6:1:1:4 [45].

The leaves and twigs of *Taxus × media* are rich in volatile oils. The volatile oil content varies, with the leaf portion being higher, making it the primary source for extraction. The volatile oil extracted from *Taxus × media* leaves and twigs contains over 20 components, with a yield ranging from 4.21% to 9.70% [27].

Using steam distillation and gas chromatography–mass spectrometry (GC–MS) to investigate the essential oils in the leaves and twigs of *Taxus × media*, 30 chemical components were identified, accounting for 97.14% of the total components. The major compounds include *cis*-3-hexen-1-ol, 3,3-dimethylacrylic acid, and pentenyl ethyl alcohol, constituting 29.42%, 18.13%, and 15.61%, respectively. The least abundant, trans-2-hexenol, accounts for only 0.24% [46]. A GC–MS analysis of essential oil components in the leaves identified 34 compounds, including benzene propanenitrile, 1,4-dioxane-2,3-diol, 3-bromo-3-methyl-butyric acid, and 1-hydroxy-2-butanone [47].

Other components present in the leaves and twigs include β-sitosterol and eicosan-10-o [38]. β-Sitosterol content is around 1% when extracted using petroleum ether and ethyl acetate [8].

The needles of *Taxus × media*, being easily regenerative and abundant in resources, yield compounds such as paclitaxel, taxol B, 10-DAB, 10β-hydroxybutyrate-10-deacetyl taxol, taxinine A, taxinine B, 5α-cinnamoyltaxicin I, and 5α-decinnamoyltaxagifine upon extraction [37,48]. The highest content is 10-DAB with approximately 1.47% and paclitaxel at about 0.308% in the extracts [49,50].

In addition to these compounds, the needles of *Taxus × media* contain a variety of flavonoid compounds, primarily flavones, dihydroflavones, and biflavones. Concentrations of specific flavonoids within the needle extracts are notably high, where myricetin 3-o-rutinoside constitutes 15.1% of the total flavonoids; quercetin 3-o-rutinoside accounts for 57.3% of the total flavonoids; kaempferol 3-o-rutinoside makes up 7.1% of the total flavonoids; quercetin 7-o-glucoside comprises 3.2% of the total flavonoids; and kaempferol 7-o-glucoside represents 0.7% of the total flavonoids. Additionally, concentrations of myricetin (0.5%), quercetin (0.4%), and kaempferol (0.4%) have also been detected within the extracts [51,52]. 7,7”-dimethoxyagastisflavone (DMGF) is a biflavone isolated from the needles of *Taxus × media*.

#### 3.1.2. Chemical Composition of *Taxus × media* Barks

The crude extract of *Taxus × media* bark contains some low-polarity substances such as chlorophyll, sterols, and resins, which can be removed with low-polarity organic solvents (like petroleum ether). Analysis of specific components in the bark and root bark of *Taxus × media* reveals that the content of paclitaxel is relatively low in these parts [3]. Apart from paclitaxel, the bark also contains 10-DAB, taxadiene, taxinine, and taxol B. Among the taxane compounds, paclitaxel is the most abundant, followed by 7-epi-10-deacetylbaccatin, cephalomannine, and 10-DAB, with paclitaxel content at about 0.0439%. The concentrations of paclitaxel and 10-DAB in the bark are higher than in the leaves and twigs [53]. These compounds also have significant medicinal value and play an important role in future drug development and research.

A study focusing on the chemical and protein components of the stem and bark of *Taxus × media* employed metabolomic and proteomic approaches. Phytochemical analysis indicated a higher concentration of paclitaxel in the phloem, and 10 critical enzymes involved in paclitaxel biosynthesis were identified, most of which are primarily produced in the phloem. Further in vitro and in vivo studies showed that TmMYB3 (*Taxus media* MYB3) participates in the biosynthesis of paclitaxel by activating the expression of taxane 2α-O-benzoyltransferase (TBT) and taxadiene synthase (TS). The phloem-specific TmMYB3 is involved in the transcriptional regulation of paclitaxel biosynthesis, potentially explaining the phloem-specific accumulation of paclitaxel [54].

#### 3.1.3. Chemical Composition of *Taxus × media* Seeds

The primary chemical components in the seeds of *Taxus × media* are taxane compounds. A variety of taxane compounds isolated from *Taxus × media* seeds include paclitaxel, taxinine A, baccatin III, 9-deacetyltaxinine, 9-deacetyltaxinine E, 2-deacetyltaxinine, taxezopidine G, 2-deacetoxytaxinine J, and 2-deacetoxytaxuspine C [3]. Seeds are also rich in flavonoid compounds, exceeding the seed coat and bark content, with substances like naringenin, aromadendrin, galanin, epigallocatechin, and gallocatechin [2].

Polyprenols have been isolated from the *Taxus × media* seeds and identified using techniques such as high-performance liquid chromatography/mass spectrometry (HPLC/MS) [55]. The results indicate that the content of TPs (taxane-based polysaccharides) in the seeds is as high as 3%, making them an alternative plant source for extracting polyprenols. Polyprenol compounds may inhibit tumor growth by inducing the cancer cells to undergo programmed cell death (apoptosis).

Recent research continues to discover and identify new compounds in different parts of *Taxus × media*, enhancing our understanding of the plant’s chemical diversity and the basis of its bioactive substances. The extraction and analysis of these compounds typically employ techniques like liquid chromatography, gas chromatography, mass spectrometry, and nuclear magnetic resonance. These techniques allow for accurate identification of compound structures, analysis of their biological activity, and exploration of their medicinal potential. As the extraction and analysis technologies advance, even more new compounds are being isolated and identified from *Taxus × media*, further expanding our understanding of the chemical diversity of this valuable plant.

### 3.2. The Impact of Origin and Growth Duration on the Variation of Active Components in Taxus × media

The variation in the active components of *Taxus × media* is influenced by multiple factors, including geographic origin, growth period, soil conditions, temperature, and humidity, all of which interact to shape the metabolic profile of the plant. Specifically, geographic origin and growth period are pivotal, significantly affecting the quantity of taxane compounds. These findings highlight how variations in the content of paclitaxel and the total amount of three key taxanes—10-DAB, cephalomannine, and baccatin III—are influenced by regional differences and growth years. Cluster analysis has revealed that the paclitaxel content in the Fuzhou region of Fujian is distinct from other areas, showing a uniquely high concentration of paclitaxel. Compared to regions such as the Jiangsu Province, Shaanxi Province, Zhejiang Province, and Yunnan Province, *Taxus × media* in the Fuzhou region exhibits significant regional specificity in its metabolite profile. This is likely due to the region’s unique subtropical climate, which provides abundant rainfall and warmth, fostering the accumulation of this metabolite. In contrast, the total amount of the three main taxanes—specifically, 10-DAB, cephalomannine, and paclitaxel—differs substantially in the Kaifeng region of Henan, which does not follow the general trend observed in other regions. Notably, the highest levels of 10-DAB and the combined levels of these three taxanes have been found in four-year-old *Taxus × media* from Kaifeng, Henan. In this case, 10-DAB levels were 3.83 times higher than in the Fujian region, cephalomannine levels in Lishui, Zhejiang, were 4.59 times higher than in Liaoning Benxi, and paclitaxel levels in Lishui, Zhejiang, were 2.17 times higher than in Liaoning Benxi. These variations are statistically significant (*p* < 0.01), emphasizing the pronounced impact of geographic origin and growth period on the biosynthesis of these crucial chemotherapeutic agents [28,56]. Moreover, *Taxus × media* can be successfully cultivated across most parts of China, with Paclitaxel content in some introduced regions even surpassing that of its original habitat [41]. 

Further studies reveal distinct concentrations of key taxane compounds when comparing the same growth durations across different regions, specifically Hainan and Sichuan. For three-year-old twigs, Hainan consistently shows higher concentrations of paclitaxel compared to Sichuan, while the 10-DAB levels are generally higher in Sichuan across multiple growth years. Cephalomannine content varies, with no consistent pattern between the two regions across the growth periods analyzed (*p* > 0.05). The distinct biochemical profiles of *Taxus × media* from these regions reflect the influence of local environmental conditions such as climate, soil type, and altitude. Hainan, characterized by its tropical monsoon climate, offers a unique environment with abundant rainfall and a longer growing season that potentially enhances the biosynthesis of certain taxanes. This environment contrasts with that of Sichuan, where the varying altitude and distinct seasonal changes might affect plant metabolism differently. These regional differences are crucial in understanding the variations in metabolite synthesis within *Taxus × media*, and provide essential insights into the optimal cultivation practices and harvest timings for maximizing the yield of these valuable compounds [57].

### 3.3. Comparison of Chemical Components between Taxus × media and Other Taxus Species

A comparative analysis of metabolites from different *Taxus* species identified 2246 metabolites. The study revealed significant differences in metabolites and compounds among various *Taxus* species. The highest contents of paclitaxel and cephalomannine are in *Taxus cuspidata*, followed by *Taxus × media*; however, the content of 10-DAB is lowest in *Taxus × media* at less than half of the other species. The highest content of baccatin III is in *Taxus yunnanensis*, slightly lower in *Taxus mairei*, while the 10-DAT content is generally low. The total content of the five taxane compounds is highest in *Taxus chinesensis*, followed by *Taxus × media*, *Taxus yunnanensis*, and *Taxus mairei* [22]. The average content of cephalomannine in the leaves of *Taxus wallichiana* is higher than in the stem, contrary to *Taxus × media*; in both species, the leaf content of paclitaxel is higher than in the stem, yet the stem and leaf content of paclitaxel in *Taxus × media* is not lower than in *Taxus wallichiana* [58].

Paclitaxel, as the most well-known component, is an effective anticancer drug, particularly significant in treating ovarian and breast cancer. Different parts of *Taxus × media* are extensively used for extracting paclitaxel [3]. Compared to other yew species, *Taxus × media* primarily excels in its significant paclitaxel content [41]. According to a report by the Sichuan Academy of Forestry Sciences, the average paclitaxel content in *Taxus × media* is 0.0385%, which is 7.3 times that of *Taxus brevifolia*, 4.6 times that of *Taxus yunnanensis*, and 4.1 times that of *Taxus mairei* [13]. In a study comparing five domestic yew species with *Taxus × media*, the branch and leaf paclitaxel content of *Taxus yunnanensis* was found to be 0.0100%, while in *Taxus × media*, it was 0.0130% [59].

## 4. Pharmacological Activity

*Taxus × media* is a plant with significant and diverse pharmacological activities, containing various compounds with distinct pharmacological properties. We will discuss its anticancer, antibacterial, anti-diabetic, anti-inflammatory, and antioxidant effects in turn. Figure 3 shows the corresponding pharmacological activities of main compounds of *Taxus × media*.

### 4.1. Anticancer Activity

#### 4.1.1. Anticancer Activity of Monomeric Compounds

##### Taxane Compounds

Paclitaxel, the principal antitumor component of *Taxus × media*, is a diterpenoid compound. Due to its concentration in the bark and leaves, it has been extensively harvested [60] and used in various traditional medical systems to treat many diseases [61,62]. Paclitaxel inhibits tumor cell mitosis and proliferation by promoting microtubule stabilization. This mechanism of action has led to its widespread use in the treatment of various types of cancer, including ovarian, breast, Kaposi’s sarcoma, and lung cancer [63]. Since its FDA approval in 1992, paclitaxel has been recognized globally as an anticancer drug [64]. Among the many taxane compounds with anticancer activity, paclitaxel is the most potent, with an IC50 value of 2.5–7.5 nM [65]. In vitro, paclitaxel has shown significant growth inhibition against transplantable tumors P388, L1210, P1534 leukemia cells, and human ovarian cancer cells, while the in vivo experiments have demonstrated its potent activity against B16 melanoma and MX1 breast cancer and activity against LX-1 lung cancer, CX-1 colon cancer, P388 leukemia, Lewis lung cancer, and Sarcoma S180 [66]. A series of hydroxylation steps of the taxane core is essential for functionalized taxane compounds, primarily mediated by cytochrome P450 enzyme-mediated oxidation. Paclitaxel is produced mainly through semi-synthesis from precursors (such as baccatin III) more readily obtained from various yew species [67]. Recent research on *Taxus × media* cell culture has revealed the potential of coronatine (COR) and coronafacic acid (CAL) to increase the production of taxanes, especially paclitaxel. The study showed that the combined use of COR and CAL significantly increased the total yield of paclitaxel and promoted its excretion into the culture medium, indicating the bi-sustainability and economic feasibility of this method in enhancing paclitaxel production [68].

Cephalomannine is primarily derived from the twigs and leaves of *Taxus × media*, where it is present in high concentrations and exhibits notable activity. Cephalomannine, found in high concentrations in the twigs and leaves of *Taxus × media*, shows strong anticancer potential, particularly against breast cancer MCF-7 cells, with a dose-dependent IC50 value of 0.86 µg/m [69]. It also effectively inhibits P388 lymphocytic leukemia, highlighting its broad-spectrum anticancer activity [70]. Cephalomannine was found to effectively inhibit the progression of bladder cancer in various experimental models, including cultured cell lines, organoids, and an in vivo model of lymphatic metastasis. Importantly, this inhibition occurred with no significant reported toxicity. These results suggest that cephalomannine, through its impact on UBE2S, holds promise as a treatment for bladder cancer, particularly in cases prone to lymphatic metastasis. This could lead to new clinical approaches for managing bladder cancer, especially for patients at high risk of metastatic disease [71]. In a previous study, male BALB/C nude mice were used as the animal model, with human lung cancer H460 cells subcutaneously implanted to simulate lung cancer. The drug used was cephalomannine, administered at a dosage of 0.4 mg/kg via intraperitoneal injection. The results indicated that cephalomannine significantly reduced tumor volume and weight. No significant loss was detected in the body and organ weights of the experimental animals, suggesting that cephalomannine significantly suppressed the growth of lung cancer cell xenografts but had no major side effects in the mice [72].

Taxinine, primarily found in the twigs and leaves of *Taxus × media*, contains various derivatives. Taxinine A exhibits cytotoxic effects on breast cancer, colon cancer, and oral squamous carcinoma cells [73]. With an IC50 value of 5.336 µg/mL, it significantly reduces MCF-7 cell proliferation after 72 h, demonstrating both time and dose dependency, albeit less potent than paclitaxel [74]. Although cephalomannine and taxinine have shown potential anticancer effects, these compounds have not yet been officially registered by the FDA or any other drug regulatory agency.

10-DAB, an effective anticancer compound, has significantly inhibited various cancer cell lines [75]. This class of compounds can inhibit the proliferation of many cancer cell lines and exert antitumor effects by inhibiting bone marrow-derived suppressor cells’ accumulation and suppressive function [76]. 10-DAB-treated MCF-7 cells, at a concentration of 5.446 µg/mL for 24 h, significantly inhibited cell proliferation with an inhibition rate of 44.8%. After 72 h of treatment, the inhibition rate was increased to 49.6% [77]. A study explored the effects of 10-DAB on tumor growth in mice infected with the Moloney murine sarcoma virus. The research utilized male NMRI mice, which were injected intramuscularly with the virus to induce tumor growth. Following the infection, the mice were treated intraperitoneally with 100 µg of 10-DAB on the first three days post infection. The results showed that while 10-DAB did not prevent tumor formation, it significantly reduced the size of the tumors compared to the control group. Specifically, 10-DAB reduced the mean diameter of the tumors by 35% compared to the control group, demonstrating its potential antitumor activity [78]. In experiments with female CDF1 mice, IDN 5390, a derivative of 10-DAB, was tested both orally and intravenously to assess its pharmacokinetic properties. Administered in doses ranging from 60 to 120 mg/kg, IDN 5390 demonstrated strong oral bioavailability at 43% for the lowest dose, though this decreased with higher doses. The drug was quickly absorbed, showing peak plasma concentrations within 15 to 30 min, and was widely distributed in vital organs like the liver, kidneys, and heart. Interestingly, its concentration in the brain remained elevated longer than in other tissues, suggesting potential utility in treating brain tumors [79]. Daily treatment with IDN 5390 in mice bearing established lung micrometastases from the B16BL6 murine melanoma caused a reduction in the size of metastases [80].

Baccatin III derivatives are precursors for the semi-synthesis of paclitaxel. Baccatin III has been found to have inhibitory effects on bleomycin A5-induced rat pulmonary fibrosis and can be used in tumor chemotherapy [81]. Additionally, it can synthesize Taxotere, a compound with higher anticancer activity. B16 melanoma shows high sensitivity to Taxotere [82]. Taxotere has shown positive effects in treating early-stage pancreatic ductal adenocarcinoma and colon adenocarcinoma, achieving multiple cures in early pancreatic ductal adenocarcinoma and colon adenocarcinoma. Taxotere also achieved an over 80% complete remission rate in the advanced stages of these two tumors [82]. In the in vitro experiments, oral administration of baccatin III significantly reduced the growth of tumors induced by engrafting BALB/c mice with either four T1 mammary carcinoma or CT26 colon cancer cells, and baccatin III decreased the accumulation of MDSCs in the spleens of the tumor-bearing mice [83].

Furthermore, 7-Xylosyl-10-deacetyltaxol shows effective inhibitory action on various tumor cell lines, with an IC50 value of 0.3776 µg/mL against the breast cancer cell line MCF-7 and an IC50 value of 0.86 µg/mL against the colon cancer cell lines, indicating its high efficacy against these cancer cell lines [84].

At present, the only taxanes are paclitaxel and the semi-synthetic drug taxotere, with baccatin III or 10-DAB as their precursor. Paclitaxel and taxotere have a wide antibacterial spectrum and are effective against a variety of drug-resistant tumor cell lines. They are mainly used in the single-drug treatment of ovarian cancer, breast cancer, small cell and non-small cell lung cancer, neck cancer, and also have significant effects on esophageal cancer, nasopharyngeal cancer, bladder cancer, lymphatic cancer, prostate cancer, malignant melanoma cancer, and gastrointestinal cancer [85].

##### Other Non-Taxane Anticancer Components

In addition to taxane compounds like paclitaxel, *Taxus × media* contains other substances with anticancer activity, such as flavonoid compounds. For instance, Apigenin exhibits significant antitumor activity [86,87]. It works through multiple mechanisms, including inducing apoptosis, regulating the cell cycle, and inhibiting cancer cell migration and invasion. Apigenin has been shown to interact with several cellular signaling pathways, such as PI3K/AKT/mTOR and MAPK/ERK [88], which are crucial in cancer treatment. Further research has also revealed the potential of 7,7”-dimethoxyagastisflavone (DMGF), extracted from *Taxus × media* cv. Hicksii, in inhibiting cancer cell proliferation. DMGF can induce apoptosis and autophagy in cancer cells and has been shown to inhibit B16F10 cell mobility in trans-well assays. Real-time PCR results indicate that DMGF also reduces the expression of matrix metalloproteinase-2 (MMP-2) and decreases vascular density of tumors in vivo [89]. Its anti-metastatic effect partly originates from the downregulation of the Cdc42/Rac1 pathway, affecting F-actin aggregation and reducing CREB phosphorylation, inhibiting pseudopodia formation [90].

Additionally, polyprenols can be isolated from the seeds of *Taxus × media*. Polyprenols are known for their various pharmacological activities, chiefly their anticancer properties. They are identified using high-performance liquid chromatography/mass spectrometry (HPLC/MS) [55]. The results indicate that the content of TPs (taxane-based polysaccharides) in the seeds is as high as 3%, making them an alternative plant source for extracting polyprenols. Polyprenol compounds may inhibit tumor growth by inducing cancer cells to undergo programmed cell death (apoptosis).

#### 4.1.2. Anticancer Activity of Extracts

The extracts of *Taxus × media* have been extensively studied and found to contain a diverse array of biologically active compounds. These include not only well-known molecules like paclitaxel but also a variety of flavonoids, terpenes, organic acids, and amino acids, all contributing to the extract’s pharmacological profile. Flavonoids and terpenes have been spotlighted for their potent anticancer properties.

Recent studies using advanced analytical techniques, such as gas chromatography–mass spectrometry (GC–MS) and high-performance liquid chromatography (HPLC), have provided deeper insights into the complex composition of these extracts. These analyses reveal the presence of multiple active components that may work in concert to exert anticancer effects, suggesting a potential synergistic interaction among these compounds. A study using gas chromatography–mass spectrometry (GC–MS) analyzed and identified compounds in the leaves of *Taxus × media*, exploring their potential biological activities. The study identified 20 compounds with significant bioactivity, mainly flavonoids, terpenes, organic acids, amino acids and their derivatives, and alcohols. Tests showed that the extracts of *Taxus × media* displayed notable anticancer properties [3]. 

The observed pharmacological activities are not solely attributable to the action of paclitaxel. While paclitaxel plays a significant role due to its well-documented anticancer efficacy, the contribution of other compounds such as the flavonoids and terpenes may enhance or complement the anticancer activity through various mechanisms [91]. For instance, some flavonoids have been shown to induce apoptosis and inhibit angiogenesis in tumor cells, while terpenes might disrupt cellular processes critical for cancer cell survival and proliferation [27].

The suggestion of a synergistic effect is particularly intriguing and warrants further investigation. Preliminary in vitro studies indicate that these extracts can inhibit the growth of various cancer cell lines more effectively than would be expected from the activity of paclitaxel alone. This synergy could be due to multiple compounds targeting different pathways involved in cancer progression, thereby increasing the overall anticancer efficacy of the extract [92].

Given these promising findings, it is crucial to pursue further clinical studies to explore the therapeutic potential of these extracts. Such studies could provide vital data on the efficacy and safety of the extracts, paving the way for their potential use as comprehensive anticancer treatments. These clinical investigations are essential to validate the anticancer activities observed in preclinical models and to assess the therapeutic viability of using *Taxus × media* extracts in oncology.

### 4.2. Antibacterial Activity

*Taxus × media* also demonstrates significant activity in combating microbial pathogens. Zhang et al. tested the antibacterial properties of essential oils extracted from fresh leaves of *Taxus × media* [47]. They used the paper disc agar diffusion method and the minimum inhibitory concentration method to assess its inhibitory effects on microbes. The results showed that the essential oil of *Taxus × media* significantly inhibits and kills microbes such as *Staphylococcus aureus* and *Escherichia coli*. This antibacterial activity is due to the combined effects of various compounds in the essential oil. The main components in the essential oil of *Taxus × media* include *cis*-3-hexen-1-ol and pentenyl ethyl alcohol, which are considered to play an essential protective role during the plant’s growth and effectively inhibit bacterial proliferation.

Additionally, compounds like benzaldehyde have been proven to have sound inhibitory effects on various microbes. Compared to the volatile oil from the leaves of *Taxus mairei*, the volatile oil from *Taxus × media* leaves show more robust antibacterial characteristics. Dar and others have extensively explored the antibacterial activity of ten different solvent extracts from the leaves against various bacteria, such as *B. pumilis*, *Staphylococcus aureus*, *Pseudomonas aeruginosa*, and *Escherichia coli*, with significant results [47,93].

Recent research has revealed the potential of endophytic fungi in medical development, especially in producing biologically active compounds. For instance, endophytic fungi isolated from *Taxus × media*, including *Graminicolous helminthosporium*, *Bipolaris australiensis*, and *Cladosporium cladosporioides*, were found to produce various bioactive compounds. These include anthraquinones, barbiturates, benzopyrroles, ethyl quinolines, etc. These endophytic fungi showed significant antifungal effects. Evaluating healthy diffusion methods for agar revealed vigorous antifungal activities in both intracellular and extracellular extracts from these fungi. Notably, endophytic fungi from Fujian Province, China, exhibited significant inhibitory capabilities against pathogenic fungi like *Neurospora* sp., *Trichoderma* sp., and *Fusarium* sp. Among them, fungi from the *Paecilomyces* sp. showed the highest positive rate of antifungal activity [47]. These substances are effective against various pathogens, including those that have developed resistance to antibiotics [94].

### 4.3. Anti-Diabetic Activity

The extracts of *Taxus × media* have shown potential in treating diabetes and its related complications in recent research. *Taxus × media* demonstrates significant anti-hyperglycemic activity [10]. A study using C57BL/6 mice fed with a high-fat diet as a model investigated the effect of *Taxus × media* extract on insulin resistance. The results indicated that the ethyl acetate extract (Tw-EA) of *Taxus × media* significantly reduced blood glucose levels, decreased the production of inflammatory cytokines, and reduced weight gain. This suggests that *Taxus × media* has therapeutic effects against inflammation-induced insulin resistance. The study also found that Tw-EA treatment reduced lipid accumulation in adipocytes and decreased the infiltration of inflammatory cells in skeletal muscle and adipose tissue, thereby improving the insulin resistance status [95].

*Taxus × media* contains paclitaxel, which is involved in some extraction and separation processes, and detection methods, and shows specific glucose-lowering effects. Its mechanism of action is different from the currently marketed diabetes treatments. It can restore the damaged pancreatic islet system in diabetic patients, potentially becoming a new direction in developing diabetes treatment drugs [96]. Dai et al. conducted a study on the glucose-lowering effects of its extract, Sequoyitol [97]. By establishing a type 2 diabetes rat model, it was found that Sequoyitol significantly reduced blood glucose levels in rats. Sequoyitol can significantly inhibit α-glucosidase activity competitively and promote glucose uptake in adipocytes, exerting a blood glucose-lowering effect comparable to 20 mg/kg of Acarbose. A further radioimmunoassay showed that Sequoyitol could reduce the insulin resistance index in rats and promote insulin secretion. p66shcA is a vital antioxidant protein [98,99], and RT-PCR studies found that Sequoyitol significantly reduced the expression of rat p66shcA mRNA even at low doses. Immunohistochemistry showed that specific doses of Sequoyitol significantly reduced the expression and phosphorylation levels of the p66shcA protein in rat thoracic aortas. Colorimetric results showed that Sequoyitol reduced the content of malondialdehyde in rat plasma. DHE staining further proved that Sequoyitol significantly inhibited the production of reactive oxygen free radicals in rat aortas [10], suggesting that Sequoyitol may benefit diabetic cardiovascular complications.

The research found that the fruit of *Taxus × media* exhibits specific anti-hyperglycemic activity. The hypoglycemic effect can be attributed to various bioactive compounds, including flavonoids, which may interact with metabolic pathways related to glucose metabolism. These interactions include affecting insulin secretion, enhancing glucose uptake in peripheral tissues, inhibiting carbohydrate-digesting enzymes, or mimicking insulin action [10].

### 4.4. Anti-Inflammatory Activity

Significant findings have been made regarding the anti-inflammatory activity of *Taxus × media*. The active component baccatin III in *Taxus × media*, known for its antitumor activity, also effectively inhibits rat pulmonary fibrosis induced by BLM. It can alleviate alveolar inflammation and the extent of pulmonary fibrosis in rats (*p* < 0.01) and reduce the expression of ERK1. Its mechanism of action is related to improving the abnormal deposition of the extracellular matrix and inhibiting excessive repair of lung injury tissues [100]. A study investigated the analgesic and anti-inflammatory activities of several compounds isolated from the bark extract, including Tasumatrol B, 1,13-Diacetyl-10-deacetylbaccatin (10-DAD), and 4-Deacetylbaccatin (4-DAB). Four hours post-administration, the 95% ethanol extract showed effective anti-inflammatory activity at 200 mg/kg concentration compared to the ether extract and the reference standard aspirin [101]. These compounds were evaluated for their potential effects on analgesia and anti-inflammation, providing further scientific evidence for the traditional medicinal use of *Taxus* species. Another study explored the anti-inflammatory action of Taxusabietane A extracted from *Taxus × media*. This study indicated that Taxusabietane A has significant anti-inflammatory activity, which aligns with its use in folk medicine for treating inflammation-related diseases. Taxusabietane A showed vigorous LOX inhibitory activity, with an IC50 value of 57 ± 0.31, and exhibited significant anti-inflammatory effects at 5 and 10 mg/kg [102].

### 4.5. Antioxidant Activity

Li et al. conducted DPPH free radical scavenging experiments on the total flavonoid extract from the twigs and leaves of *Taxus × media* [43]. Comparing the scavenging rates of various concentration gradients of the extract with Vitamin C on DPPH free radicals, it was found that its antioxidant activity increases with the rise in mass concentration. The scavenging rates for DPPH free radicals, ABTS^+^ free radicals, and nitrite were 91.04%, 99.17%, and 65.50%, respectively, indicating that the total flavonoids of *Taxus × media* twigs and leaves possess strong in vitro antioxidant capabilities.

## 5. Discussion and Conclusions

*Taxus × media*, a valuable plant with medicinal, timber, and ornamental significance and originally from North America, is now widely cultivated globally with successful introductions in regions like Sichuan, China. This plant plays a crucial role in various ecosystems, and its unique growth environment and ecological characteristics are vital for developing effective cultivation and conservation strategies. However, facing resource depletion and increased environmental pressures, this species confronts multiple challenges, including habitat loss, overharvesting, and the impacts of climate change. In the past thirty years, many subpopulations, especially in China, have experienced a decline of over 30%. In countries like India, Nepal, and Vietnam, their conservation status has reached critical and endangered levels [103,104]. Therefore, understanding its cultivation status, assessing the survival of wild populations, and exploring effective conservation strategies are crucial for its protection. The successful cultivation of *Taxus × media* is essential for ecological protection and the development of medical resources.

Regarding its phytochemical components, the rich bioactive substances in *Taxus × media,* including paclitaxel and its derivatives, various alkaloids, and flavonoid compounds, are the primary sources of its medicinal value. Taxanes are important bioactive components but most of their contents seem to be very low and even deficient for some taxanes (Table 1). Current research focuses include the extraction, preparation, analysis, and biological activity assessment of these compounds. Exploring the structure and function of these chemical components is crucial for uncovering the material basis of *Taxus × media* and developing new drugs. However, a comprehensive analysis of the related literature on *Taxus × media* indicates the need for more data on the content of these components in the bark and seeds, including many other practical components, necessitating further research. Future studies should focus on the detailed distribution of these critical components in different parts of *Taxus × media*, which is essential for a better understanding of its material basis and the development of medical resources. Additionally, there is a pressing need to study how the controlled factors in artificial plantation settings—such as watering, nutrition, pest control—affect the metabolite content and seasonal variation of these compounds. Therefore, future studies need to fill this gap by exploring the distribution of these critical components in different parts of *Taxus × media*, which is essential for a better understanding of its material basis and the development of medical resources.

Pharmacological activity is a significant aspect of *Taxus × media*. Studies have found that *Taxus × media*’s potential is not limited to its anticancer effects but further extends to applications in anti-diabetes, anti-inflammatory, antimicrobial, and other fields. Identified taxane and non-taxane components may explain why the extracts of *Taxus × media* have anticancer activities or others. These studies reveal the action and mechanisms of related compounds in *Taxus × media*, providing a theoretical basis for new drug development and indicating a new direction for further clinical applications. However, since the extracts contain many kinds of taxane compounds, whether those taxane compounds have similar molecular action mechanisms to paclitaxel remains unclear. Also, aside from anticancer activities, efficiently discovering new activities of the *Taxus × media* compounds and revealing their mechanisms is an important future research direction.

The use of plant extracts containing cytotoxins such as paclitaxel in alternative medicine is an important aspect that deserves attention. Although paclitaxel is a well-established chemotherapeutic agent approved for conventional medical use, its incorporation into alternative medical practices raises significant safety concerns. Paclitaxel possesses potent cytotoxic properties, capable of killing cancer cells at very low concentrations, which also implies potential toxicity to healthy cells if not administered with precise control. This duality underscores the need for careful evaluation and regulation when considering such powerful compounds for use in non-traditional treatments. Our discussion aims to highlight the challenges and responsibilities involved in leveraging such potent pharmacological agents outside of controlled medical settings. If *Taxus* extracts were clinically used, a necessary preclinical study and strict population or patient screening should be conducted in advance to avoid poisoning accidents. Further research is crucial for developing safer, modified derivatives of these compounds that retain therapeutic efficacy while reducing toxicity, making them more suitable for widespread use in alternative therapies. Advanced extraction methods should be developed to enhance the effectiveness and reduce the toxicity of *Taxus*-derived compounds. Although the extract contains toxic compounds, e.g., paclitaxel, and should be used with caution, other components in the extract may enhance the efficacy and reduce toxicity of paclitaxel, which is unique in alternative medicine and may happen in the extract of *Taxus*. However, further validation research is needed. Additionally, in alternative medicine, combining these with other herbs may increase the effectiveness and safety of *Taxus* formulations. In fact, specific formulas have already been developed in China, but research into more effective and safer *Taxus* formulations continues.

Internationally, *Taxus × media* and other Asian yew species are listed in Appendix II of the Convention on International Trade in Endangered Species of Wild Fauna and Flora (CITES) to regulate their international trade. This species enjoys first-class protection in China and appears in essential nature reserves, such as the Tangjiahe National Nature Reserve in Sichuan. Moreover, to ensure future sustainable harvests, China and other countries like Nepal, Bhutan, and Vietnam have invested significantly in establishing plantations. However, due to the slow growth of yew plants and their rarity, the insufficient supply of raw materials for paclitaxel production has become an urgent issue. Current cutting-edge techniques can rapidly expand the area of artificial cultivation. New chemical synthesis techniques for semi-synthesizing intermediates like 10-DAB from *Taxus × media* can partially solve the problem of relatively low paclitaxel content in the plant.

Additionally, techniques for producing paclitaxel and its precursors through endophytic fungi or in vitro cell cultures are being developed, which could partially address the dependency on yeast cultivation. However, these technologies still need to mature and are costly. Therefore, finding new plant resources and developing new technologies to increase paclitaxel production is one of the current challenges. 

As we advance our understanding of *Taxus × media*, it becomes imperative to integrate climate resilience into conservation strategies. The susceptibility of *Taxus* to increased temperatures might reduce their natural range, necessitating the exploration of adaptive cultivation techniques that can withstand broader environmental variations. Furthermore, research into genetic variations across different *Taxus* populations could identify strains with higher resilience to climate stressors, thereby guiding conservation and breeding programs to focus on these more robust specimens.

In summary, as a plant rich in bioactive compounds, research on the origin and cultivation of *Taxus × media*, its phytochemical components, and pharmacological activity is of great significance for new drug development and biodiversity conservation. The development of yew is a systemic project involving multiple disciplines and fields, including agricultural cultivation, environmental protection, active component extraction and identification, biological activity research, and medical applications. Strengthening the protection of this precious species, formulating and implementing more effective conservation measures, and achieving sustainable development will be vital to ensuring its biodiversity and ecological balance. Notably, future research should focus on paclitaxel and the comprehensive utilization of its various components in different medical fields, warranting further anticipation and attention. 

## Figures and Tables

**Figure 1 ijms-25-05756-f001:**
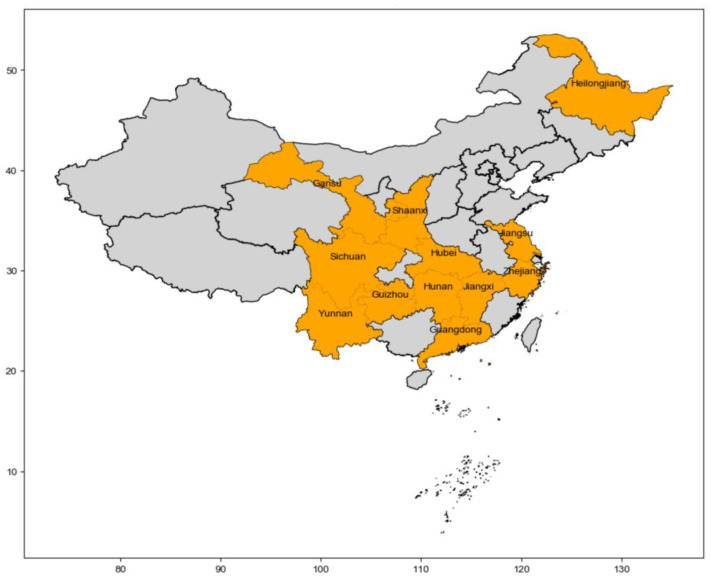
The geographical distribution of *Taxus × media* in China.

**Figure 2 ijms-25-05756-f002:**
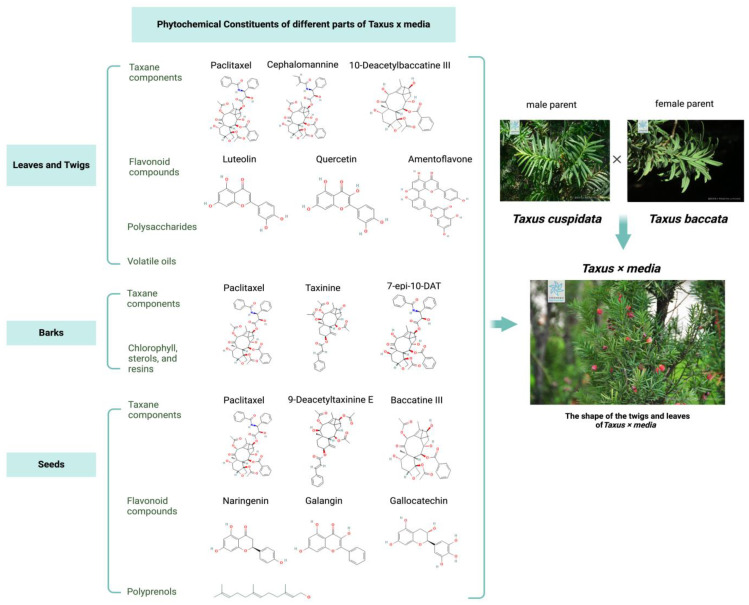
The phytochemical constituents of the different parts of *Taxus × media*.

**Figure 3 ijms-25-05756-f003:**
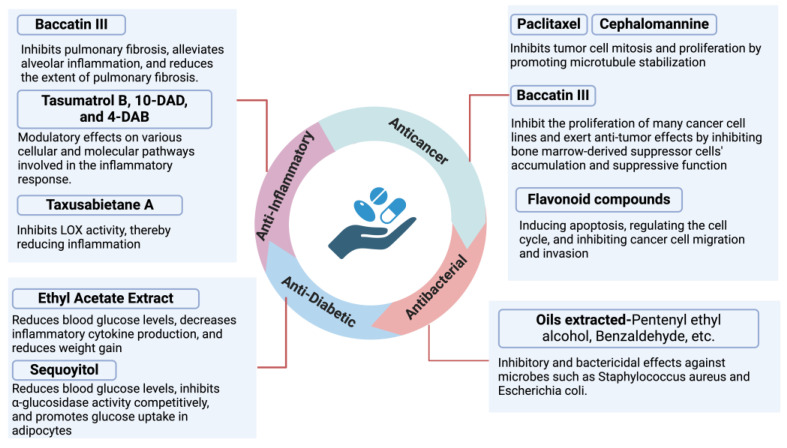
Corresponding pharmacological activities of main compounds of *Taxus × media*.

**Table 1 ijms-25-05756-t001:** Main chemical constituents and contents of *Taxus × media*.

Bioactive Constituents	Molecular Weight	CAS	Chemical Structure	Source	Content	Reference
Paclitaxel	853.9 g/mol	33069-62-4	C47H51NO14	The leaves, needles and barks of *Taxus × media*	0.13–0.60 mg/g	[2,26,27]
Cephalomannine	831.9 g/mol	71610-00-9	C5H53NO14	The leaves, needles and barks of *Taxus × media*	0. 053 mg/g	[2]
Baccatine III	586.6 g/mol	27548-93-2	C31H38O11	The needles of *Taxus × media*	0.13–0.31 mg/g	[26]
10-DAB	544.6 g/mol	32981-86-5	C29H36O10	The leaves and needles of *Taxus × media*	0.24–0.78 mg/g	[28,29]
Taxinine	606.7 g/mol	3835-52-7	C35H42O9	The needles and bark of *Taxus × media*		[30]
Taxinine A	476.6 g/mol	18530-09-1	C26H36O8	The seeds and needles of *Taxus × media*		[30,31]
9-Deacetyltaxinine E	608.7 g/mol	284672-78-2	C35H44O9	The seeds of *Taxus × media*		[32]
10-DAT	544.6 g/mol	32981-86-5	C29H36O10	The needles of *Taxus × media*	0. 126 mg/g	[33]
7-epi-10-DAT	544.6 g/mol	7162-920	C29H36O10	The needles of *Taxus × media*	0.033 mg/g	[2]
7-Xylosyl-10-deacetyltaxol	944.0 g/mol	90332-63-1	C50H57NO17	The needles of *Taxus × media*	0.315 mg/g	[34,35]

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
