# Peer review of "Research on the Medicinal Chemistry and Pharmacology of Taxus × media"

_ijms, 2024, doi:10.3390/ijms25115756_

Round 1

Reviewer 1 Report

Comments and Suggestions for Authors

I found the reviewed topic interesting considering the pharmacological and clinical importance of taxane derivatives. To me, the fact of having some species and varieties of Taxus with higher content of bioactive metabolites is quite important to maintain a constant natural supply of such drugs. Also, the artificial cultivation of this genus is very interesting. Nonetheless, I consider, several sections of the review are described very general, some points are not discussed enough, and there were not perspectives about important issues. Here, I wrote little mistakes and some questions I would like you to answer and add to the manuscript.  

Lines 39 -40: “With the increasing depletion of natural resources and environmental pressures, Taxus ×media faces numerous challenges, including habitat loss, overharvesting, and the impact of climate change”.

I found this very interesting and outstanding topic, but you did not discuss about it or give any comment about what are the perspective of the survival and conservation of Taxus before the climate change? Are Taxus specimens susceptible to temperature increase or other factors? How can them endanger the conservation of this genus? And so on. Please add this to the manuscript.

Line 239. “Several factors can influence the variation of…”

Please mention what are those factors and how they influence the metabolite variation of Taxus. If this review aims to be a guide for further research, then I would to have the whole information and do not have the need to go to the original source to look by myself.

Lines 241-243: “Research indicates that, apart from the lack of a significant correlation between 10-Deacetylbaccatin III and Cephalomannine, there is a highly significant positive correlation among the rest of the components (P<0.01)”

Again, mention what are those other components how they behave. Are they correlated with age, origin, or both? What kind of components are they? Also, are they taxanes or other type of metabolites? Do they vary the equally or differentially?  

In line 243 (P<0.01) letter “P” must be small p and italicized. The same for al “P” in the manuscript.

Lines 247: “Cluster analysis has shown that the Paclitaxel content in the Fuzhou region of Fujian has a certain uniqueness. At the same time, the total number of three taxanes in Kaifeng, Henan, differs from other regions”

Please, mention how was such variation, what region possessed the highest and lowest contents? Are they significant differences?

Lines 250 – 253: “Another study shows that under different growth durations, the highest contents of 10-Deacetylbaccatin III and Paclitaxel in 3-year-old twigs, and the highest content of Cephalomannine in 7-year-old twigs are found in Taxus × media cultivated in Hainan.”

First, you never mention what are the other regions compared with Hainan. Second, you never mention what make Hainan geographical region different from the other regions, which could be implicated in its higher content of metabolites. This not even discussed later in the manuscript. Please add such information and discussion in in appropriated sections.

Lines 510 – 512: “However, a comprehensive analysis of related literature on Taxus ×media indicates a need for more data on the content of these components in the bark and seeds, including many other practical components, necessitating further research.”

What are those other many components needing to be studied? I consider this part is when you provide your point of view of what is missing and needing to be studied about this topic. I mean, this would be a contribution of your review by guiding other researchers according to your revisions what should be attended to help in the use and conservation of Taxus. So please, add what you think is necessary, in addition to organ metabolite distribution, to be further studied. For instance, factors that can be controlled in artificial plantation of taxus such as watering, nutrition, pest control, and how these factors might influence metabolite content and seasonal variation.

 In fact, I would like you to add information describing the features of artificial cultivation systems of Taxus. Moreover, add information about how similar or different, in terms of metabolites and some morphological features, are wild and cultivated taxus specimens. If there are or not such differences what repercussions or advantages represent of the exploitation of Taxus metabolite obtention.  Also, if there are no studies about the influence of climate or geographical factors on the chemical variation of taxus, you could add information about related families and try to discus if probably this might be the same case for Taxus.

Other general corrections:

1)      Chemical or metabolite names should be always written in small letter, unless the metabolite name is the first word of the sentence. So please, change all metabolites names in small letters in the whole manuscript when they are not at the beginning of a sentence.

2)      Prefixes such as cis-, trans-, o-, m-, or p- must be italicized in chemical names.

3)      ABTS+ in line 483 must be written with superscript plus symbol.

4)      At line 475 you wrote IC(50) before you wrote it as IC50, please chose one and keep an homogeneous format in the text.

5)      At lines 405 – 406 al bacteria names must be italicized. Indeed, all names in section 4.2.

6)      Letter size in figure 1, specifically provinces’ names are too small and not well distinguished.

7)      The words pharmacological activity in figure 3 are no needed in the figure or at least you mention pharmacological activity of what or whose.

Comments on the Quality of English Language

There are some issues with english such as not propper election of some wrod in some sentences or the use of really small paragrags which are not even conected to the revious one. For example, the ones at page number 5.

Author Response

Dear Reviewer,

Thank you for giving us an opportunity to revise our manuscript, we appreciate you very much for your positive and constructive comments and suggestions on our manuscript entitled “Research on the Medicinal Chemistry and Pharmacology of Taxus × media” (ID: ijms-2962993).

According to your comments, we have made necessary modifications to our manuscript and added additional explanations to make our results convincing. The revised paragraphs (sentences) are labeled in different colors.

We would like to express our great appreciation to you for comments on our paper. Looking forward to hearing from you.

Best regards.

Xinyu Gao

Weidong Xie

Tel: +86-755-26036086

Reviewer 2 Report

Comments and Suggestions for Authors

There are two fundamental drawbacks in this paper.

1. A book on Paclitaxel by Kumaraswamy, M, Pullaiah, T. and Zhe-Seng Chen has been published in 2011 by Elsevier. Much of the information given in this paper is available in this book. Authors deliberately ignored this book.

2. Scientific names are not in italics. Highlighted in the text.

Author Response

(The authors gave the same response as above.)

Reviewer 3 Report

Comments and Suggestions for Authors

An outstanding example of missing criticism is paragraph 4.1 Anticancer activity of monomeric compounds. Most of this paragraph describes the cytotoxicity of the compounds. Cancer is a disease meaning that anticancer compounds are compounds that can cure cancer. Most of the compounds mentioned here are cytotoxic, but to be named as anticancer compounds they should show selectivity against cancer cells. This is known for paclitaxel but not for most of the other compounds mentioned. Thus, the authors need to either relate to animals or even better clinical experiments to confirm their statement. In line 286 is claimed that extracts possess pharmacological activities. Since paclitaxel is present this is not surprising. 

Comments on the Quality of English Language

See other comments.

Author Response

(The authors gave the same response as above.)

Reviewer 4 Report

Comments and Suggestions for Authors

The manuscript „Research on the Medicinal Chemistry and Pharmacology of Taxus × media” raises interesting issues. The quality of this manuscript is very poor. The manuscript is ill-conceived, there are many repetitions and editorial errors, which significantly reduces its quality. In its current form, I suggest rejecting this manuscript.

Line 43, 50, 60, 115, 226, 235, 380, 392, 400, 437, 479 – reference

Line 68-69 – verbal notation of numerals, please use them according to the spelling rules

Latin names of species - italics – all manuscript

„within the Taxus genus of the Taxaceae family” - repetitions

Line 90- Natural habitant?

 “Taxus × media” “in vitro” “in vivo”– italic

Figure 2. Photo of Taxus is not visible

Why do you write chemical compounds with capital letters?

P-hydroxybenzoic acid „p” from „para” please check the correctness of writing all chemical compounds

The total flavonoid extraction from 156 Taxus × media leaves and twigs is about 128.1mg/g[28] -expressed in…

What is the difference between Taxus needle and leaves?

Line 307, 325, 328, 337, 338…: units written incorrectly

Line 516: wrong place for table caption

Author Response

(The authors gave the same response as above.)

Round 2

Reviewer 2 Report

Comments and Suggestions for Authors

Scientific names are not in italics at many places.

In references all the scientific names are not in italics.

These are highlighted.

They should be in italics 

Author Response

(The authors gave the same response as above.)

Reviewer 3 Report

Comments and Suggestions for Authors

In my previous evaluation of the manuscript, I asked some questions to the authors. I expected answers to these questions but have only gotten a general thank from the authors for my work. Instead, the authors have made a new manuscript but they have not commented on my questions.

I see no Fig. 3

I miss a clear statement about which compounds have been registered as drugs by FDA or other authorities. Paclitaxel is registered as taxol but what about cephalomannine and taxinine?

I also questioned the use of plant extracts containing cytotoxins like paclitaxel in alternative medicines. This problem has not been mentioned.

No structures are given for some of the compounds mentioned in Table 1.

Comments on the Quality of English Language

No comments.

Author Response

(The authors gave the same response as above.)

Round 3

Reviewer 2 Report

Comments and Suggestions for Authors

Revised version is fine.